# Chemical Constituents of the Egg Cases of *Tenodera angustipennis* (Mantidis ootheca) with Intracellular Reactive Oxygen Species Scavenging Activity

**DOI:** 10.3390/biom11040556

**Published:** 2021-04-10

**Authors:** Seung Mok Ryu, Hyeon-hwa Nam, Joong Sun Kim, Jun-ho Song, Young Hye Seo, Hyo Seon Kim, A Yeong Lee, Wook Jin Kim, Dongho Lee, Byeong Cheol Moon, Jun Lee

**Affiliations:** 1Herbal Medicine Resources Research Center, Korea Institute of Oriental Medicine (KIOM), Naju 58245, Korea; smryu@kiom.re.kr (S.M.R.); hhnam@kiom.re.kr (H.-h.N.); centraline@kiom.re.kr (J.S.K.); songjh@kiom.re.kr (J.-h.S.); wnsl1118@kiom.re.kr (Y.H.S.); hs0320@kiom.re.kr (H.S.K.); lay7709@kiom.re.kr (A.Y.L.); ukgene@kiom.re.kr (W.J.K.); bcmoon@kiom.re.kr (B.C.M.); 2Department of Plant Biotechnology, College of Life Sciences and Biotechnology, Korea University, Seoul 02841, Korea; dongholee@korea.ac.kr

**Keywords:** Mantidis ootheca, *Tenodera angustipennis*, *N*-Acetyldopamine derivative, tenoderin, antioxidant activity

## Abstract

As a traditional medicine with potential antioxidant effects, *Tenodera angustipennis* egg cases (Mantidis ootheca) are a potential source of new bioactive substances. Herein, three new *N*-acetyldopamine derivatives, namely, (+)-tenoderin A (**1a**), (−)-tenoderin A (**1b**), and tenoderin B (**2**), along with thirteen known compounds (**3**–**15**), were isolated from a 70% EtOH extract of *T*. *angustipennis* egg cases. Compound **1** was isolated as a racemic mixture, and two enantiomers (**1a** and **1b**) were successfully separated by chiral-phase preparative HPLC. The chemical structures of the new compounds were established by NMR spectroscopy and high-resolution electrospray ionization mass spectrometry, and the absolute configurations of enantiomers **1a** and **1b** were determined by electronic circular dichroism spectroscopy. All the new compounds exhibited antioxidant activities with IC_50_ values of 19.45–81.98 μM, as evaluated using free-radical scavenging assays, with the highest activity observed for compound **2**. In addition, compounds **1a**, **1b**, and **2** exhibited inhibitory activities on intracellular reactive oxygen species generation.

## 1. Introduction

Mantidis ootheca, which refers to egg cases of the Mantidae family, such as *Tenodera angustipennis* Saussure, *Hierodula patellifera* Serville, *Statilia maculate* Thunberg, and *Tenodera sinensis* Saussure, has been used as a source of traditional medicines in East Asia, including Korea and China [1]. Mantidis ootheca has been used in traditional medicines to treat incontinence, lumbago, spermatorrhea, acidosis, renal failure, and leukorrhea [2], and has been reported to have various biological activities, including anti-inflammatory, antidiuretic, anticancer, and antioxidant activities, as well as vascular relaxant effects [3,4,5,6]. Although various pharmacological studies on Mantidis ootheca have been reported, only a few studies have investigated its chemical constituents. The chemical constituents of Mantidis ootheca have been revealed to include fatty acids and phenols that exhibit pharmacological properties, such as antibacterial, antibiofilm, antioxidant, and anti-atherosclerotic effects [7,8,9]. Notably, previous studies have indicated that commercially available Mantidis ootheca products contain the egg cases of several mantis species within one package [10]. As such mixtures may be used for research, the exact identification of research samples is of particular importance.

As part of our continuing research on the discovery of new bioactive substances in Mantidis ootheca [10,11], the egg cases of *T. angustipennis*, which constitute a major raw material within Mantidis ootheca product mixtures, were selected for this study through sample identification. Using a 70% EtOH extract of *T*. *angustipennis* egg cases, which was selected as an antioxidant resource through our screening system, three new *N*-acetyldopamine derivatives (**1a**, **1b**, and **2**) and thirteen known compounds (**3–15**) were isolated. Compound **1** was isolated as an enantiomeric mixture, and two stereoisomers (**1a** and **1b**) were successfully separated by chiral-phase preparative HPLC. Their chemical structures were identified using spectroscopic/spectrometric techniques including NMR spectroscopy, electronic circular dichroism (ECD) spectroscopy, and high-resolution electrospray ionization mass spectrometry (HRESIMS). To assess the antioxidant activities of the isolated compounds, their free-radical 1,1-diphenyl-2-picrylhydrazyl (DPPH) and 2,2′-azino-bis-3-ethylbenzthiazoline-6-sulfonic acid (ABTS) scavenging activities and their inhibitory activity on intracellular reactive oxygen species (ROS) generation were evaluated.

## 2. Materials and Methods

### 2.1. General Experimental Procedures

UV, optical rotation, and ECD spectra were recorded using a Pop UV-vis spectrophotometer (Mecasys, Daejeon, Korea), a P-2000 polarimeter (Jasco, Tokyo, Japan), and a J-1100 spectrometer (Jasco), respectively. LC/MS and HRESIMS data were acquired using a UPLC Q-TOF MS spectrometer (Waters, Milford, MA, USA). GC/MS data were recorded using an Agilent 6890/JMS-700 system (Agilent, Palo Alto, CA, USA). NMR data were acquired using a 500 MHz NMR spectrometer (Bruker, Karlsruhe, Germany). MPLC was performed using a Biotage Selekt system (Biotage AB, Uppsala, Sweden). Preparative HPLC was performed using a Waters system with an YMC-Pack ODS-A column (5 µm, 250 × 20 mm I.D.) and a Chiralpak IH column (5 µm, 250 × 4.6 mm I.D.).

### 2.2. Insect Material

The insect material (Mantidis ootheca) was purchased from a medicinal herb company (Gwangmyeongdang Co., Ulsan, Korea) in July 2019, authenticated, and deposited in the Korean Herbarium of Standard Herbal Resources (Index Herbarium code KIOM, specimen no. 2-20-0223) at the Korea Institute of Oriental Medicine. To prepare the sample for this study, the egg cases of *T*. *angustipennis* were identified and selected based on key morphological characteristics, such as shape, color, texture, and angle of the distal end [10]. All materials had a fusiform shape, a tapered distal end with angles of 22°–50°, and a lusterless and brownish textured external wall (Appendix A).

### 2.3. Extraction and Isolation

Dried *T*. *angustipennis* egg cases (900.0 g) were ground and extracted with 70% EtOH (3 × 10.0 L) at room temperature. After the solvent was evaporated, the residue was suspended in distilled water (4.0 L) and extracted using EtOAc (3 × 4.0 L) to obtain the EtOAc-soluble extract (23.6 g). The EtOAc-soluble extract (20.0 g) was fractionated by MPLC using Diaion HP20 (Mitsubishi Chemical, Tokyo, Japan) cartridges (400 g, H_2_O–MeOH, 100:0 to 0:100 in 18 CV, 40 mL/min) to produce six fractions (Fr 1–6). Fr 2 (1368.3 mg) was fractionated by MPLC using Sfär Silica HC (Biotage AB, Uppsala, Sweden) cartridges (200 g, *n*-hexane–EtOAc, 80:20 to 0:100 in 30 CV, 30 mL/min) to produce nine sub-fractions (Fr 2.1–2.9). Fr 2.2 (3.7 mg) was purified using preparative HPLC (H_2_O–ACN, 80:20 to 40:60 in 50 min, 6 mL/min) to yield **5** (1.2 mg). Fr 2.4 (44.3 mg) was purified using preparative HPLC (H_2_O–ACN, 85:15 to 70:30 in 50 min, 6 mL/min) to yield **6** (23.4 mg). Fr 2.5 (43.6 mg) was purified using preparative HPLC (H_2_O–ACN, 85:15 to 70:30 in 50 min, 6 mL/min) to yield **10** (14.4 mg). Fr 2.6 (34.9 mg) was purified using preparative HPLC (H_2_O–ACN, 85:15 to 70:30 in 50 min, 6 mL/min) to yield **9** (15.5 mg), **11** (3.2 mg), **12** (1.8 mg), and **13** (2.4 mg). Fr 2.9 (1648.3 mg) was fractionated by MPLC using Sephadex LH_20_ (Pharmacia Fine Chemicals Inc., Piscataway, NJ, USA) cartridges (120 g, H_2_O–MeOH, 100:0 to 0:100 in 16 CV, 20 mL/min) to produce nine sub-fractions (Fr 2.9.1–2.9.9). Fr 2.9.3 (114.6 mg) was purified using preparative HPLC (H_2_O–ACN, 90:10 to 80:20 in 50 min, 6 mL/min) to yield **3** (39.8 mg) and **4** (40.1 mg). Fr 2.9.5 (55.1 mg) was purified using preparative HPLC (H_2_O–ACN, 85:15 to 75:25 in 50 min, 6 mL/min) to yield **1** (6.3 mg). Racemic mixture **1** was separated by preparative HPLC (*n*-hexane–EtOH–MeOH, 70:20:10 isocratic in 15 min, 1 mL/min) with a chiral-phase stationary column (Chiralpak IH) to yield **1a** (2.0 mg, *t*_R_ 4.8 min) and **1b** (2.1 mg, *t*_R_ 11.7 min). Fr 4 (863.9 mg) was fractionated by MPLC using Sfär Silica HC cartridges (100 g, CHCl_3_–MeOH, 100:0 to 50:50 in 40 CV, 70 mL/min) to produce seven sub-fractions (Fr 4.1–4.7). Fr 4.1 (53.5 mg) was purified using preparative HPLC (H_2_O–ACN, 80:20 to 60:40 in 50 min, 6 mL/min) to yield **7** (6.9 mg) and **8** (11.2 mg). Fr 4.4 (198.5 mg) was purified using preparative HPLC (H_2_O–ACN, 80:20 to 70:30 in 50 min, 6 mL/min) to yield **2** (14.6 mg). Fr 5 (2176.2 mg) was fractionated by MPLC using Sfär Silica HC cartridges (200 g, CHCl_3_–MeOH, 100:0 to 50:50 in 21 CV, 80 mL/min) to produce five sub-fractions (Fr 5.1–5.5). Fr 5.3 (473.7 mg) was fractionated by MPLC using Sephadex LH_20_ cartridges (120 g, H_2_O–MeOH, 100:0 to 0:100 in 17 CV, 30 mL/min) to produce nine sub-fractions (Fr 5.3.1–5.3.7). Fr 5.3.2 (32.7 mg) was purified using preparative HPLC (H_2_O–ACN, 75:25 to 65:35 in 50 min, 6 mL/min) to yield **14** (1.6 mg). Fr 6 (9810.0 mg) was purified by MPLC using Sfär Silica HC cartridges (350 g, *n*-hexane–CHCl_3_–MeOH, 100:0:0 to 50:50:0, 0:100:0 to 0:0:100 in 20 CV, 50 mL/min) to yield **15** (2012.2 mg).

#### 2.3.1. Tenoderin A (**1**)

Brown gum; ^1^H and ^13^C NMR (500 and 125 MHz, CD_3_OD), see Table 1; ESIMS (positive) *m/z* 403 [M + H]^+^; ESIMS (negative) *m/z* 401 [M − H]^−^; HRESIMS *m/z* 401.1349 [M − H]^−^ (calcd for C_20_H_21_N_2_O_7_, 401.1349).

##### (+)-tenoderin A (**1a**)

Brown gum; [α]^26^_D_ +21.2 (*c* 0.01, MeOH); UV (MeOH) λ_max_ (log *ε*) 215 (3.91), 285 (3.37), 332 (3.23) nm; ECD (*c* 0.5 mM, MeOH) Δ*ε* +33.9 (212), −19.5 (248), +20.1 (282), +7.6 (302), +11.3 (317).

##### (−)-tenoderin A (1b)

Brown gum; [α]^26^_D_ −21.0 (*c* 0.01, MeOH); UV (MeOH) λ_max_ (log *ε*) 215 (3.91), 285 (3.36), 332 (3.25) nm; ECD (*c* 0.5 mM, MeOH) Δ*ε* −36.6 (212), +8.1 (247), −25.1 (284), −12.6 (302), −13.1 (317).

#### 2.3.2. Tenoderin B (**2**)

Brown gum; UV (MeOH) λ_max_ (log *ε*) 215 (3.95), 285 (3.91) nm; ^1^H and ^13^C NMR (500 and 125 MHz, CD_3_OD), see Table 1; ESIMS (positive) *m/z* 316 [M + H]^+^; ESIMS (negative) *m/z* 314 [M − H]^−^; HRESIMS *m/z* 314.1042 [M − H]^−^ (calcd for C_17_H_16_NO_5_, 314.1028).

### 2.4. Computational Methods

ECD calculations were performed for **1a** and **1b**. A conformer distribution was constructed using the Spartan’14 software (Wavefunction, Inc., Irvine, CA, USA) with a Merck molecular force field. The conformers were optimized at the DFT [B3LYP/6-31+G(d,p)] level, and ECD calculations were performed at the TDDFT (CAM-B3LYP/SVP) level with a CPCM solvent model in MeOH using the Gaussian 09 software (Gaussian, Inc., Wallingford, CT, USA).

### 2.5. DPPH and ABTS Radical Scavenging Activities

The DPPH (Sigma-Aldrich, St. Louis, MO, USA) and ABTS (Sigma-Aldrich) free-radical scavenging activities were measured according to previous studies [12,13]. All measurements were independently repeated three times. The DPPH and ABTS scavenging activities were calculated as follows: DPPH or ABTS scavenging activity (%) = [A(free radical) − A(standard)]/A(free radical), where A is the absorbance value of DPPH at 517 nm or ABTS at 734 nm.

### 2.6. Detection of Intracellular ROS

The HUVECs used in the ROS scavenging experiments were purchased from the American Type Culture Collection (ATCC, Manassas, VA, USA). The cells were cultured in α-MEM medium, and 20% FBS in a humid atmosphere of 5% CO_2_ at 37 °C.

To confirm the antioxidant effect of each compound against oxidative stress, the cells were treated with each sample at a concentration of 5, 10, 50 or 100 µM and hydrogen peroxide (1 mM), cultured for 24 h, washed with PBS, and treated with 100 μM DCFDA (ThermoFisher, Waltham, MA, USA) for 30 min at 37 °C. Reacted. After DCFDA was removed, and the amount of DCFDA remaining in the cells was observed using a fluorescence microscope. The fluorescence intensity at an excitation wavelength of 485 nm and an emission wavelength of 520 nm was measured using a microplate reader.

## 3. Results and Discussion

Dried *T*. *angustipennis* egg cases selected from a commercial Mantidis ootheca material were ground and extracted with 70% EtOH. Following fractionation, fifteen compounds were isolated from the extract and structural determination was performed (Figure 1).

Compound **1** was obtained as a brown gum, and HRESIMS analysis revealed its elemental formula to be C_20_H_22_N_2_O_7_, suggesting eleven degrees of unsaturation. The ^1^H NMR data indicated the presence of two methyl groups (*δ*_H_ 1.80 (3H, s, CH_3_-10) and 2.02 (3H, s, CH_3_-10′)), two methylene groups (*δ*_H_ 2.55 (2H, t, *J* = 7.2 Hz, H-7) and 3.24 (2H, t, *J* = 7.2 Hz, H-8)), a methine group (*δ*_H_ 6.66 (1H, s, H-8′)), two *meta*-coupled aromatic methine groups (*δ*_H_ 6.61 (1H, d, *J* = 1.8 Hz, H-2) and 6.39 (1H, d, *J* = 1.9 Hz, H-6)), and three aromatic methine groups in an ABX spin system (*δ*_H_ 6.72 (1H, d, *J* = 8.4 Hz, H-5′), 7.44 (1H, d, *J* = 2.0 Hz, H-2′), and 7.47 (1H, dd, *J* = 8.4, 2.1 Hz, H-6′)) (Table 1). The ^13^C NMR data corresponded to 20 carbons, including two methyl carbons, two methylene carbons, six methine carbons, three carbonyl carbons, and seven additional quaternary carbons (Table 1). The ^1^H and ^13^C NMR data for compound **1** were similar to those for the dimeric *N*-acetyldopamine derivative polyrhadopamine B [14], except that the 2-oxo-*N*-acetyldopamine (**4**) unit linked at C-6 in polyrhadopamine B was linked at C-5 in compound **1**. This result was supported by the presence of two doublet proton signals with meta coupling at H-2 (*δ*_H_ 6.61) and H-6 (*δ*_H_ 6.39), as well as the HMBC correlations between H-6/C-8′ (*δ*_C_ 54.4), C-7 (*δ*_C_ 36.0), C-2 (*δ*_C_ 116.8), and C-4 (*δ*_C_ 142.8) and those between H-8′/C-6 (*δ*_C_ 120.3), C-4 (*δ*_C_ 142.8), C-5 (*δ*_C_ 125.3), C-9′ (*δ*_C_ 173.2), and C-7′ (*δ*_C_ 196.3) (Figure 2).

Compound **1** is a racemic mixture, which was suggested by the lack of an optical rotation value. Therefore, chiral separation was performed on a Chiralpak IH column to yield optically pure enantiomers **1a** (+21.2 (*c* 0.01, MeOH)) and **1b** (−21.0 (*c* 0.01, MeOH)) (Figure 3). The absolute configurations of **1a** and **1b** were determined by comparing the calculated and experimental ECD spectra. The calculated ECD spectra of the 8′*R* and 8′*S* stereoisomers were in good agreement with the experimental spectra of **1a** and **1b**, respectively (Figure 4). Therefore, the structures of new compounds **1a** and **1b** were assigned as (+)-tenoderin A and (−)-tenoderin A, respectively.

Compound **2** was obtained as a brown gum, and its elemental formula was determined to be C_17_H_17_NO_5_ using HRESIMS analysis. The 1D NMR data revealed the presence of a *N*-acetyldopamine (**3**) moiety as well as a 1,4-disubstituted benzene ring (*δ*_H_ 7.68 (2H, d, *J* = 8.2 Hz, CH_3_-2′ and 6′) and 6.84 (2H, d, *J* = 8.3 Hz, CH_3_-3′ and 5′); *δ*_C_ 134.3 (C-2′ and 6′) and 116.2 (C-3′ and 5′)) and a ketone group (*δ*_C_ 199.1 (C-7′)) (Table 1). A detailed analysis of the NMR data showed that compound **2** was similar to periplanetol A [15], except for the absence of a hydroxy group at C-5′ and the presence of an aromatic methine signal (*δ*_H_ 6.84, H-5′) for compound **2**. These results are supported by the COSY correlations between H-5′/H-6′ and the HMBC correlations between H-5′/ C-4′ (*δ*_C_ 164.0), C-3′ (*δ*_C_ 116.2), and C-1′ (*δ*_C_ 131.3) (Figure 2). Accordingly, the structure of new compound **2** was elucidated, and this compound was given the trivial name tenoderin B.

In addition to the above-described new compounds, thirteen known compounds (**3–15**) were isolated and identified as *N*-acetyldopamine (**3**), 2-oxo-*N*-acetyldopamine (**4**) [16], 4-hydroxybenzaldehyde (**5**) [17], 4-hydroxybenzoic acid (**6**) [18], apocynin (**7**) [19], benzoic acid (**8**) [20], protocatechuic acid (**9**) [21], 4-hydroxyphenylacetic acid (**10**) [22], (*S*)-1-phenylenthane-1,2-diol (**11**) [23], 4-hydroxyphenylglyoxylic acid amide (**12**) [24], (±)-hydroxybutenolide (**13**) [25], scoparone (**14**) [26], and oleic acid (**15**) (Figure 1). Notably, many of the known compounds (**4–14**) were first isolated from Mantidis ootheca.

The antioxidant activity of the extract, as measured using DPPH and ABTS radical scavenging assays, was significant (81.99 ± 1.98% and 99.74% ± 0.13%, respectively, at 100 μg/mL) (Appendix A). Furthermore, all the isolated compounds were screened for antioxidant activity at a concentration of 100 μM using DPPH and ABTS radical scavenging assays. DPPH and ABTS radical scavenging activities were observed for compounds **1a**, **1b**, **2**, **3**, **4**, and **9** (Appendix A). In particular, new compounds **1a**, **1b**, and **2** displayed antioxidant activities with IC_50_ values between 19.45 and 81.98 μM. Among the new compounds, the antioxidant activity of compound **2** was the highest. In addition, although compounds **1a** and **1b** were enantiomers, there was a difference in their antioxidant activities (Table 2). Compound **1b** exhibited stronger antioxidant effects than **1a**, and the ratios of their IC_50_ values were 1:1.8 (DPPH) and 1:1.3 (ABTS). These results suggest that the difference in antioxidant activity between the two enantiomers is due to the chirality of C-8′.

In addition, to determine whether the antioxidant activities were associated with protective effects in H_2_O_2_-treated human umbilical vein/vascular endothelium cells (HUVECs), intracellular 2′,7′-dichlorofluorescin diacetate (DCFDA) levels were measured. Oxidative-stress-induced cell damage has been implicated in various types of disease. H_2_O_2_ is a major ROS produced intracellularly during pathological processes and causes oxidative injury. H_2_O_2_ has been extensively used as an inducer of oxidative stress for in vitro models [27]. Therefore, H_2_O_2_ was selected to promote oxidative stress in the current investigation. In this study, as shown in Figure 5, H_2_O_2_ significantly increased the intracellular DCFDA level. In this assay system, compound **2** showed a potent antioxidant activity in a dose-dependent manner and compounds **1a** and **1b** also showed significant antioxidant activity, which presented the identical activity pattern supporting the results in the free-radical scavenging assay.

## 4. Conclusions

In conclusion, this study on the chemical constituents of *T*. *angustipennis* egg cases (Mantidis ootheca) revealed sixteen compounds, including three new *N*-acetyldopamine derivatives (**1a**, **1b**, and **2**) and thirteen known compounds (**3–15**). Two enantiomers (**1a** and **1b**) were successfully separated by chiral-phase preparative HPLC, and the absolute configurations were determined by ECD spectroscopy. All the isolated compounds were evaluated for antioxidant activity, and the new compounds appeared to show antioxidant effects in HUVECs. These findings not only reveal various chemical constituents in the egg cases of *T*. *angustipennis,* they also provide guidance for clarifying the pharmacodynamic basis of the antioxidant effects of Mantidis ootheca.

## Figures and Tables

**Figure 1 biomolecules-11-00556-f001:**
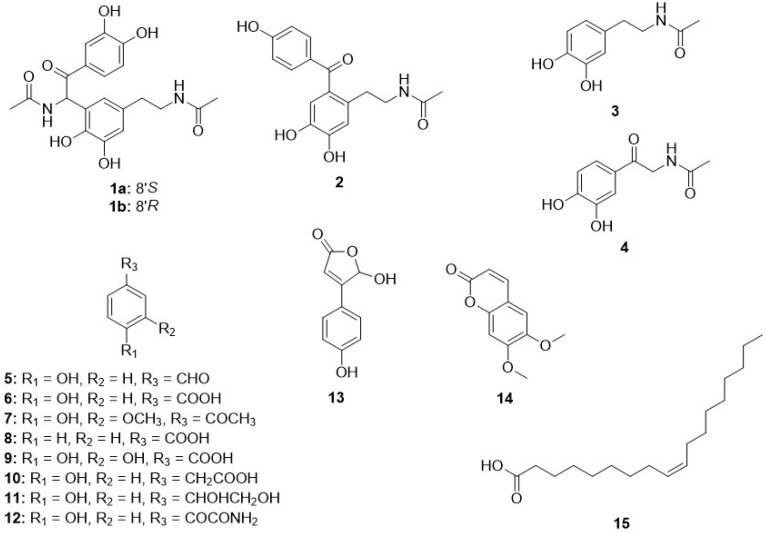
All isolated compounds from a 70% EtOH extract of *T*. *angustipennis* egg cases.

**Figure 2 biomolecules-11-00556-f002:**
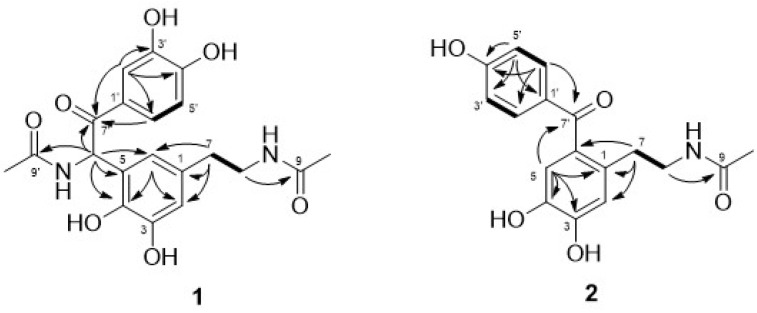
Key HMBC (arrow) and COSY (bold) correlations (**1** and **2**).

**Figure 3 biomolecules-11-00556-f003:**
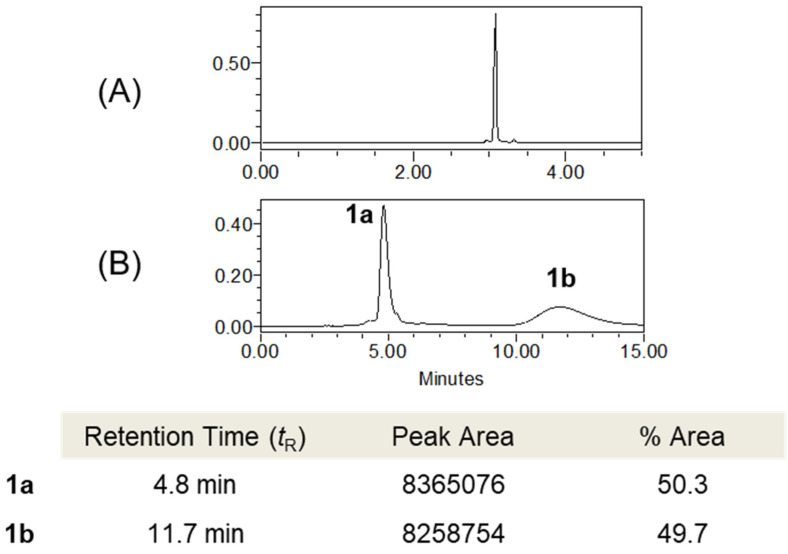
(**A**) UPLC chromatogram of compound **1** (stationary phase: ACQUITY UPLC BEH C_18_ column (2.1 mm × 100 mm, 1.7 μm); mobile phase: H_2_O–ACN, 95:5 to 50:50 in 5 min, 0.3 mL/min; UV 280 nm). (**B**) HPLC chromatogram of the enantiomeric mixture of compound **1** (stationary phase: Chiralpak IH column (4.6 mm × 250 mm, 5 μm); mobile phase: *n*-hexane–EtOH–MeOH, 70:20:10 isocratic in 15 min, 1 mL/min; UV 280 nm).

**Figure 4 biomolecules-11-00556-f004:**
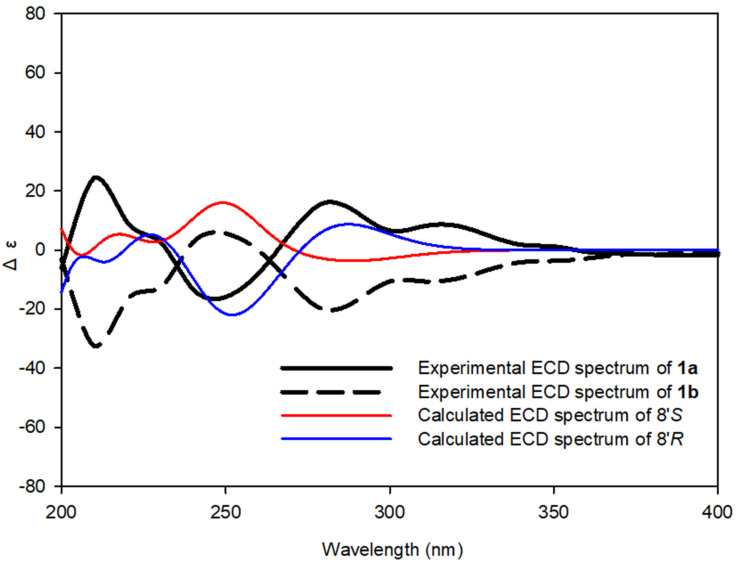
Calculated and experimental electronic circular dichroism (ECD) spectra of **1a** and **1b**.

**Figure 5 biomolecules-11-00556-f005:**
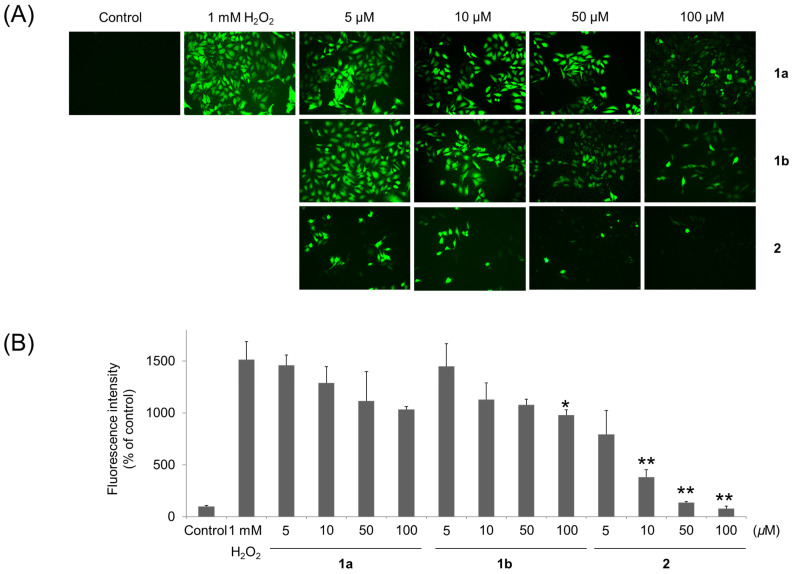
(**A**) Reactive oxygen species (ROS) scavenging activities as revealed by images of human umbilical vein/vascular endothelium cells (HUVECs) with 2′,7′-dichlorofluorescin diacetate (DCFDA) treated with 5–100 μM of compound **1a**, **1b**, or **2**. (**B**) Scavenging percentages of the tested compounds. * *p* < 0.05, ** *p* < 0.01.

**Table 1 biomolecules-11-00556-t001:** 1D NMR (^1^H and ^13^C) and 2D NMR (HMBC and COSY) data for compounds **1** and **2** in CD_3_OD.

Position	1				2			
	*δ*_C_, Type ^2^	*δ*_H_, Multi (*J* in Hz) ^1^	COSY	HMBC	*δ*_C_, Type	*δ*_H_, Multi (*J* in Hz)	COSY	HMBC
1	132.4, C				132.5, C			
2	116.8, CH	6.61, d (1.8)		3, 4, 6, 7	118.7, CH	6.78, s		3, 4, 6, 7
3	147.4, C				149.1, C			
4	142.8, C				144.1, C			
5	125.3, C				118.1, CH	6.75, s		1, 3, 4, 7′
6	120.3, CH	6.39, d (1.9)		2, 4, 7, 8′	131.7, C			
7	36.0, CH_2_	2.55, t (7.2)	8	1, 2, 6, 8	33.1, CH_2_	2.73, t (7.0)	8	1, 2, 6, 8
8	42.2, CH_2_	3.24, t (7.2)	7	1, 7, 9	42.8, CH_2_	3.30, overlap ^3^	7	1, 7, 9
9	173.4, CO				173.4, CO			
10	22.6, CH_3_	1.80, s		9	22.7, CH_3_	1.83, s		9
1′	128.5, C				131.3, C			
2′	116.7, CH	7.44, d (2.0)		3′, 4′, 6′, 7′	134.3, CH	7.68, d (8.2)	3′	3′, 4′, 6′, 7′
3′	146.5, C				116.2, CH	6.84, d (8.3)	2′	1′, 4′, 5′
4′	152.6, C				164.0, C			
5′	115.9, CH	6.72, d (8.4)	6′	1′, 3′, 4′	116.2, CH	6.84, d (8.3)	6′	1′, 3′, 4′
6′	123.9, CH	7.47, dd (8.4, 2.1)	5′	2′, 4′, 7′	134.3, CH	7.68, d (8.2)	5′	2′, 4′, 5′, 7′
7′	196.3, CO				199.1, CO			
8′	54.4, CH	6.66, s		4, 5, 6, 7′, 9′				
9′	173.2, CO							
10′	22.4, CH_3_	2.02, s		9′				

^1^ Measured at 500 MHz. ^2^ Measured at 125 MHz. ^3^ Overlap with NMR solvent (CD_3_OD).

**Table 2 biomolecules-11-00556-t002:** IC_50_ values for the antioxidant activities of new compounds **1a**, **1b**, and **2**.

Samples	IC_50_ (μM)	
	DPPH	ABTS
1a	81.50 ± 0.77 ^1^	81.98 ± 0.48
1b	46.54 ± 0.56	62.74 ± 0.69
2	19.45 ± 0.42	37.23 ± 0.26
Gallic acid	8.95 ± 0.20	10.82 ± 0.97

^1^ Values are reported as mean ± SD (n = 3).

## Data Availability

Not applicable.

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
