# Peer review of "Chemical Constituents of the Egg Cases of *Tenodera angustipennis* (Mantidis ootheca) with Intracellular Reactive Oxygen Species Scavenging Activity"

_biomolecules, 2021, doi:10.3390/biom11040556_

Round 1
Reviewer 1 Report
The manuscript reports the identification of 16 compounds including three new compounds from the 70% EtOH extract of T. angustipennis egg cases, the chemical constituents of which are rarely studied previously. Particularly, compounds 1a and 1b are racemic mixture, which was further separated by chiral HPLC and their absolute configuration was determined by ECD calculation. The antioxidant activities of all the isolated compounds were evaluated using DPPH and ABTS radical scavenging assays, and the new compound 3 displayed the most potent antioxidant activity in a dose-dependent manner. The manuscript is well written and the NMR and MS data fully support the structures elucidated. I recommend the acceptance of this manuscript if some minor issues as follows can be solved.
- Page 7. “The antioxidant activity of the extract, as measured using DPPH and ABTS radical scavenging assays, was high (81.27% ± 4.42% and 99.74% ± 0.13%, respectively, at 100 µg/mL)”. Does the word “high” here mean “potent”? It sounds unreasonable that the antioxidant activity of the extract is potent if the IC50 value is larger than 80 µM.
- Page 7. The intracellular ROS was measured in H2O2-treated human umbilical vein/vascular endothelium cells upon the treatment with compounds. The authors should give more detailed description and discussion of the results in the manuscript.
Author Response
The response to the Reviewer #1’s comments:
The manuscript reports the identification of 16 compounds including three new compounds from the 70% EtOH extract of T. angustipennis egg cases, the chemical constituents of which are rarely studied previously. Particularly, compounds 1a and 1b are racemic mixture, which was further separated by chiral HPLC and their absolute configuration was determined by ECD calculation. The antioxidant activities of all the isolated compounds were evaluated using DPPH and ABTS radical scavenging assays, and the new compound 3 displayed the most potent antioxidant activity in a dose-dependent manner. The manuscript is well written and the NMR and MS data fully support the structures elucidated. I recommend the acceptance of this manuscript if some minor issues as follows can be solved.
1.1. Page 7. “The antioxidant activity of the extract, as measured using DPPH and ABTS radical scavenging assays, was high (81.27% ± 4.42% and 99.74% ± 0.13%, respectively, at 100 µg/mL)”. Does the word “high” here mean “potent”? It sounds unreasonable that the antioxidant activity of the extract is potent if the IC50 value is larger than 80 µM.
→ Answer: We fully agree your comment. We have changed “high” to “significant” in manuscript (page 7, lines 226).
1.2. Page 7. The intracellular ROS was measured in H2O2-treated human umbilical vein/vascular endothelium cells upon the treatment with compounds. The authors should give more detailed description and discussion of the results in the manuscript.
→ Answer: As suggested, we have added more detailed description and discussion in “3. Results and Discussion” section (page 8, lines 243-248 and 249-252).

Reviewer 2 Report
The manuscript entitled "Chemical constituents of the egg cases of Tenodera angustipennis..." reports the structure and activity against ROS of three new compounds.
The manuscript was remarkably well prepared.
Please consider the following remarks.
line 183. The way of presenting 2D NMR data you chose in Fig. 2 is the one commonly adopted in most of publications reporting new natural products. However, the concept of "key correlation" is not really well defined. You may supplement Table 1 with all HMBC and COSY correlations (HSQC is implicit).
You also may deposit your raw NMR data (I use personally the zenodo.org web site for this purpose) so that anyone interested in examining your data in detail would be able to do it, according to FAIR (Findable, Accessible, Interoperable, Reusable) principles. The spectra in Supplementary data file are generally only useful for a vague recognition of spectral features by eye and cannot be further exploited.
line 186. "suggested" would be more appropriate than "evidenced". Chiral compounds may cause a negligible optical rotation, by accident.
line 222. "81.27% ± 4.42%" Are you sure that so many significant digits are really significant? Taking into account all the potential source of errors, two significant digits would be a maximum. 81.27% ± 4.42%" refers to some value between 77% and 86%.
line 229. "...there was a difference in their antioxidant activities". Any comment about this observation?
Author Response
The response to the Reviewer #2’s comments:
The manuscript entitled "Chemical constituents of the egg cases of Tenodera angustipennis..." reports the structure and activity against ROS of three new compounds.
The manuscript was remarkably well prepared.
Please consider the following remarks.
2.1. Line 183. The way of presenting 2D NMR data you chose in Fig. 2 is the one commonly adopted in most of publications reporting new natural products. However, the concept of "key correlation" is not really well defined. You may supplement Table 1 with all HMBC and COSY correlations (HSQC is implicit).
→ Answer: We have added HMBC and COSY correlations data in Table 1 (page 3).
2.2. You also may deposit your raw NMR data (I use personally the zenodo.org web site for this purpose) so that anyone interested in examining your data in detail would be able to do it, according to FAIR (Findable, Accessible, Interoperable, Reusable) principles. The spectra in Supplementary data file are generally only useful for a vague recognition of spectral features by eye and cannot be further exploited.
→ Answer: Thank you for your suggestion. Since our NMR data contain information on new substances, we will upload them after the paper is published.
2.3. Line 186. "suggested" would be more appropriate than "evidenced". Chiral compounds may cause a negligible optical rotation, by accident.
→ Answer: We appreciate your comment. We have changed “evidenced” to “suggested” in manuscript (page 6, lines 190).
2.4. Line 222. "81.27% ± 4.42%" Are you sure that so many significant digits are really significant? Taking into account all the potential source of errors, two significant digits would be a maximum. 81.27% ± 4.42%" refers to some value between 77% and 86%.
→ Answer: We fully agree your comment and admit for our mistakes. DPPH scavenging rate (absorbance) of extracts was measured at 83.07% (0.086), 83.19% (0.0854), and 79.70% (0.1031) in 100 ug/mL. As a result of reconfirming through the SPSS 12.0 K. statistical program, it was confirmed as 81.99 ± 1.98%. A value between 77.06% and 86.90% was measured as a significant value. Therefore, we have corrected SD value "81.27% ± 4.42%" to "81.99 ± 1.98%" in manuscript (page 7, lines 226) and supporting information S17. Furthermore, we have confirmed that other SD data is clear.
2.5. Line 229. "...there was a difference in their antioxidant activities". Any comment about this observation?
→ Answer: As suggested, we have added more detailed description and discussion in “3. Results and Discussion” section (page 7, lines 234-237).
